# An Update of Evidence for Pathogen Transmission by Ticks of the Genus Hyalomma

**DOI:** 10.3390/pathogens12040513

**Published:** 2023-03-25

**Authors:** Sarah I. Bonnet, Stéphane Bertagnoli, Alessandra Falchi, Julie Figoni, Johanna Fite, Thierry Hoch, Elsa Quillery, Sara Moutailler, Alice Raffetin, Magalie René-Martellet, Gwenaël Vourc’h, Laurence Vial

**Affiliations:** 1Ecology and Emergence of Arthropod-borne Pathogens unit, Institut Pasteur, Université Paris-cité, Centre National de Recherche Scientifique (CNRS) UMR 2000, Institut National de Recherche pour l’Agriculture, l’Alimentation et l’Environnement (INRAE) USC 1510, 75015 Paris, France; 2Interactions Hôtes-Agents Pathogènes unit, Université de Toulouse, Institut National de Recherche pour l’Agriculture, l’Alimentation et l’Environnement (INRAE), Ecole Nationale Vétérinaire de Toulouse (ENVT), F-31076 Toulouse, France; 3Bioscope Corse Méditerranée unit UR7310, Faculté de Sciences, Campus Grimaldi, Université de Corse, 20250 Corte, France; 4Santé publique France, 94410 Saint-Maurice, France; 5Risk Assessment Department, French Agency for Food, Environmental and Occupational Health & Safety (ANSES), 94700 Maisons-Alfort, France; 6Biologie Épidémiologie et Analyse de Risque en santé animale unit, Oniris, Institut National de Recherche pour l’Agriculture, l’Alimentation et l’Environnement (INRAE), 44300 Nantes, France; 7Biologie Moléculaire et immunologie Parasitaire unit, Laboratoire de Santé Animale, Agence nationale de sécurité sanitaire de l’alimentation, de l’environnement et du travail (ANSES), Institut National de Recherche pour l’Agriculture, l’Alimentation et l’Environnement (INRAE), Ecole Nationale Vétérinaire d’Alfort (ENVA), 94700 Maisons-Alfort, France; 8Reference Centre for Tick-Borne Diseases, Paris and Northern Region, Department of Infectious Diseases, General Hospital of Villeneuve-Saint-Georges, 94190 Villeneuve-Saint-Georges, France; 9Epidémiologie des maladies animales et zoonotiques unit, Université de Lyon, Institut National de Recherche pour l’Agriculture, l’Alimentation et l’Environnement (INRAE), VetAgro Sup, 69280 Marcy l’Etoile, France; 10Epidémiologie des maladies animales et zoonotiques unit, Université Clermont Auvergne, Institut National de Recherche pour l’Agriculture, l’Alimentation et l’Environnement (INRAE), VetAgro Sup, 63122 Saint-Genès-Champanelle, France; 11Animal Santé Territoires Risques Ecosystèmes unit, Centre de coopération international en recherche agronomique pour le développement (CIRAD), Institut National de Recherche pour l’Agriculture, l’Alimentation et l’Environnement (INRAE), Université Montpellier II, F-34398 Montpellier, France

**Keywords:** ticks, *Hyalomma* sp., tick-borne pathogens, vectorial competence

## Abstract

Current and likely future changes in the geographic distribution of ticks belonging to the genus *Hyalomma* are of concern, as these ticks are believed to be vectors of many pathogens responsible for human and animal diseases. However, we have observed that for many pathogens there are no vector competence experiments, and that the level of evidence provided by the scientific literature is often not sufficient to validate the transmission of a specific pathogen by a specific *Hyalomma* species. We therefore carried out a bibliographical study to collate the validation evidence for the transmission of parasitic, viral, or bacterial pathogens by *Hyalomma* spp. ticks. Our results show that there are very few validated cases of pathogen transmission by *Hyalomma* tick species.

In addition to their direct impact as ectoparasites, ticks are of greater importance as vectors of pathogens to animals and are the second most important group of pathogen vectors affecting humans after mosquitoes, mainly due to their transmission of *Borrelia burgdorferi* sensu lato [1]. In addition, these arthropods are capable of transmitting the largest variety of pathogens including bacteria, parasites (protozoa, helminths), and viruses. However, like all biological vectors, ticks are not simple “syringes”, as is often believed. Each species, or even each tick population, has a vector competence corresponding to its intrinsic ability to acquire the pathogen by feeding on an infected host, allowing the multiplication/development of this agent and its retransmission to a new host during a new blood meal [2]. To this vector competence is added the set of factors that may influence transmission, thus defining the vector capacity—that is, the ability of a vector to transmit a pathogen at a given time and in a defined region, according to extrinsic conditions such as humidity, temperature, but also vector abundance, trophic preferences, etc. [3]. 

Ticks acquire pathogens during a blood meal taken on infected vertebrate hosts. Several routes of pathogen transmission to vertebrate hosts are then possible on the occasion of a new blood meal via a deposit on the skin of the host and penetration via the bite wound (via feces, by crushing or via coxal fluid excretion), or via the injection of saliva that accompanies the blood meal and represents the predominant route of pathogen transmission for ticks [4]. In addition, within the tick population, a pathogen may persist from one life stage to the next via transstadial transmission (essential for tick-borne transmission by hard ticks that take only one blood meal per life stage), from the female to its offspring via transovarial transmission, from male to female ticks during copulation via sexual transmission, or from an infected tick to a non-infected one via co-feeding when ticks feed adjacent to each other on the same host [2,5].

The unique detection of pathogenic DNA in a tick collected in the environment or on a vertebrate host does not prove vector competence, but only indicates the fact that the tick took a blood meal on an infected animal. Although such DNA detection in unfed hard ticks is more indicative than detection in engorged ones—as, considering that ixodid ticks feed only once per life stage, it suggests transstadial persistence—it should be noted that this does not imply the viability of the concerned pathogen. In fact, the persistence of DNA during the molting process is possible, as detection has been reported of some vertebrate DNA dating back to a blood meal taken by the previous life stage [6]. Furthermore, the DNA of pathogens that are not transmitted by the tick species concerned is often detected simply as a result of feeding on hosts that harbor such pathogens [7]. The detection of mRNA, a priori reflecting the presence of a living organism, presents an additional but still insufficient level of evidence of vectorial transmission. Additionally, the detection of a given pathogen in both ticks and samples from tick-infested vertebrate hosts collected in the same area, the co-occurrence with already known tick-borne pathogens, or a documented infection following a tick bite, can all represent indirect significant evidence of a given pathogen transmission. The demonstration of transstadial and/or transovarial persistence, which validates the existence of a development of the pathogen in ticks, is a strong indication in favor of biological vector transmission. However, conclusive evidence of vector competence for a given pathogen can only be provided by the demonstration of the ability of a tick species to acquire a pathogen on an infected host, to allow its development, and to transmit it to a new host. Unfortunately, very few vector competence experiments have been conducted due to the difficulties encountered in carrying out complete transmission cycles under experimental conditions. Indeed, it requires having pathogen-free tick colonies, vertebrate hosts suitable for both tick engorgement and pathogen replication (or to have effective artificial tick feeding methods combined with optimal cultivation methods of the pathogen), and can require high biosecurity levels depending on the pathogen concerned [8,9].

Ticks from the *Hyalomma* genus are considered to be expanding from some parts of their range, as reported for the invasion of H. marginatum into Europe since the late 20th century [10,11]. This is of concern, as these ticks are vectors of many pathogens responsible for human and animal diseases [12] and because there are few measures to control them, in particular during their off-host development [13]. Like other hard ticks, *Hyalomma* species take one blood meal per life stage before molting (larva and nymph) or, after fertilization, laying eggs (female). The majority of *Hyalomma* spp. ticks have a three-host cycle (each of the three life stages must find a new host on which to take a blood meal). However, some are diphasic, such as those of the marginatum group (larvae and nymphs taking their blood meals on the same host), and one species, *Hyalomma scupense*, is monophasic (all three life stages remain on the same host). *Hyalomma* ticks feed on domestic or wild vertebrate hosts, with some species such as H. marginatum or H. rufipes utilizing a large variety of hosts, which favors pathogen spillover. Humans, by entering the ecosystem of these hosts, can become accidental hosts of ticks, and thus, become exposed to pathogens [14]. The genus *Hyalomma* includes the most xerophilous species among all ticks and may be favored under future climate change [15].

Numerous pathogens—parasitic, viral, or bacterial—have been reported in the scientific literature as transmitted or potentially transmitted by ticks of the genus Hyalomma. The synthesis of these studies, with an attributed level of evidence of vectorial transmission, is reported in Table 1. The level of evidence ranges from a simple detection of pathogen DNA or RNA in ticks collected from vertebrate hosts to a formal demonstration of experimental transmission from an infected vertebrate host to a new naïve one, coupled with appropriate epidemiological data. To build this table, we considered the 27 Hyalomma species described by Guglielmone et al. in 2010 [16]. Species for which no evidence of a potential vector role has been reported to date have not been included, namely *Hyalomma albiparmatum, Hyalomma arabica, Hyalomma brevipunctatum, Hyalomma glabrum, Hyalomma hystricis, Hyalomma nitidum, Hyalomma punt, Hyalomma rhipicephaloides, Hyalomma franchinii*, and *Hyalomma kumari*. Our literature review includes the names of Hyalomma species that have been used for several past decades but have since been abandoned in favor of the currently used names, namely *Hyalomma plumbeum* (now *H. marginatum*) and *Hyalomma detritum* (now *H. scupense*). Note that the data identified for *Hyalomma savignyi*, now reclassified as *H. lusitanicum*, are not considered here, since *H. savignyi* is now considered to include several subspecies. *Hyalomma savignyi* data could therefore also apply to *H. lusitanicum*, as well as to *H. anatolicum*, *H. impeltatum*, *H. impressum*, *H. marginatum*, or *H. truncatum*. For bibliographic research, a narrative review was performed using the terms “*Hyalomma*” and “[pathogen sought]” (all microorganisms whose transmission by ticks has been reported in the scientific literature with de facto exclusion of symbionts) with the Boolean operator “AND” in the PubMed and Scopus databases without date restriction. The literature search was conducted in English. We retained peer-reviewed research articles and reviews (not including conference proceedings) and book sections. Screening was conducted first on titles, then on abstracts, and finally on the full text when available. After reading the entire articles, the ones that were eliminated corresponded to those that did not have available data or no original data. The number of references found in each of the two databases is shown in Table 1. All references concerning experimental validation and the epidemiological arguments for transmission are mentioned in the table, but the list concerning DNA/RNA detection is not exhaustive. 

In conclusion, we observed that there are many missing pathogen vector competence experiments, and that the level of evidence provided by the scientific literature is often not sufficient to validate the existence of vectorial transmission. We conclude that the pathogen/tick associations for which transmission from an infected host to an initially healthy host via tick bite has been experimentally validated are the following: Crimean–Congo Hemorrhagic Fever Virus (CCHFv) by *H. dromedarii*, *H. impeltatum*, *H. marginatum*, *H. rufipes*, and *H. truncatum*.African Horse Sickness virus by *H. dromedarii*.Venezuelan equine encephalitis virus by *H. truncatum*.*Theileria annulata* by *H. anatolicum*, *H. dromedarii*, *H. excavatum*, *H. lusitanicum*, and *H. scupense*.*Theileria equi* by *H. anatolicum* and *H. excavatum*.*Theileria lestoquardi* by *H. anatolicum*.*Theileria ovis* by *H. anatolicum*.*Babesia occultans* by *H. rufipes*.*Coxiella burnetii* by *H. aegyptium*.*Anaplasma marginale* by *H. excavatum*.*Rickettsia aeschlimannii* by *H. marginatum* and *H. rufipes*.

## Figures and Tables

**Table 1 pathogens-12-00513-t001:** Bibliographic review of evidence in favor of the transmission of pathogens by ticks of the Hyalomma genus.

Tick Species	Number of References in PubMed/Scopus	Pathogen Transmitted/Suspected to Be Transmitted	Detection of DNA/RNA/Antigen/Pathogen in Ticks	Epidemiological Arguments of Possible Transmission *	Experimental Validation of Transmission **	References
** *H. aegyptium* **	96/110	CCHFv	RNA	yes	0	[17]
		*Coxiella burnetii*	DNA	no	2, 4	[18,19]
		*Borrelia turcica* *Borrelia* spp.	DNA and RNA	no	2	[20,21,22,23,24,25]
		*Bartonella bovis*	DNA	no	0	[26]
		*Ehrlichia canis* *Ehrlichia* spp.	DNA	yes	0	[18,25,26,27]
		*Anaplasma phagocytophilum*	DNA	yes	0	[18]
		*Rickettsia africae*	DNA	no	0	[28]
		*Rickettsia aeschlimannii*	DNA	yes	0	[26,29,30]
		*Rickettsia sibirica mongolitimonae*	DNA	no	0	[30,31]
		*Rickettsia slovaca*	DNA	no	0	[30]
		Meram virus	RNA	no	0	[32]
		Tamdy virus	RNA	no	0	[32]
** *H. anatolicum* **	381/546	*Babesia caballi*	DNA	no	0	[33]
		*Theileria equi*	DNA, pathogen	yes	2, 4	[33,34,35,36,37]
		*Babesia occultans*	DNA	no	0	[38,39,40]
		*Babesia bovis*	DNA	no	0	[38,39,40,41]
		*Theileria annulata*	DNA, pathogen	yes	2, 4	[35,39,41,42,43,44,45,46,47,48,49,50,51]
		*Theileria lestoquardi*	DNA	yes	2, 4	[35,44,49,52,53,54,55,56]
		*Theileria ovis*	DNA	yes	2, 4	[33,35,38,40,57,58]
		*Babesia ovis*	DNA	no	0	[55]
		CCHFv	RNA, antigen, viral particle	uncertain	1	[59,60,61,62,63,64,65]
		Alphavirus	RNA	no	0	[66]
		Zahedan Rhabdovirus	RNA	no	0	[67]
		Tick Borne Encephalitis virus	no	no	5	[68]
		Kadam virus	RNA	no	0	[66]
		Karshi virus	no	no	0, 1	[69]
		Karyana virus	RNA, virus isolation	yes	0	[70]
		Kundal virus	RNA, virus isolation	yes	0	[70]
		Sindbis virus	RNA	no	0	[71]
		*Coxiella burnetii*	DNA	no	0	[72,73,74]
		*Bartonella* spp.	DNA	no	0	[38]
		*Borrelia* spp.	DNA	no	0	[38]
		*Anaplasma marginale, Anaplasma phagocytophilum, Anaplasma ovis, Anaplasma centrale, Ehrlichia* spp., *Rickettsia massiliae, Rickettsia* spp.,	DNA	no	0	[38,41,75]
*H. asiaticum*	145/192	*Theileria annulata*	DNA	no	0	[76,77,78]
		*Babesia occultans*	DNA	no	0	[79]
		*Babesia caballi*	DNA	no	0	[80,81]
		*Theileria equi*	DNA	no	0	[80]
		CCHFv	RNA, viral particles	yes	1	[65,82,83,84,85]
		Chim virus	RNA	no	0	[86]
		Syr-Darya valley fever virus	RNA	no	0	[86]
		Karshi virus	RNA	no	0, 1	[69,87,88]
		Tamdy virus	Virus isolation	yes	0	[89,90,91,92]
		*Coxiella burnetii*	DNA	no	2	[74,93,94,95]
		*Rickettsia siberica*	DNA	no	0	[96]
		*Borrelia burgdorferi* s.l.	RNA	no	0	[97]
		*Rickettsia sibirica mongolitimonae*	isolation	yes	0	[98]
*H. dromedarii*	232/344	*Theileria equi*	DNA, pathogen in ticks	no	1, 2, 4	[99,100,101,102]
		*Theileria camelensis*	Pathogen in ticks	yes	1, 2, 4	[103,104,105]
		*Theileria annulata*	DNA	yes	2, 4	[48,106,107,108,109,110,111,112]
		*Theileria ovis*	DNA	no	0	[40]
		*Babesia caballi*	DNA	no	0	[101]
		*Babesia occultans*	DNA	no	0	[101]
		CCHFv	RNA, antigen, viral particles	yes	1, 2, 4	[64,113,114]
		Alphavirus	RNA	no	0	[66]
		Chick Ross virus	RNA	no	0	[66]
		Dera Ghazi Khan virus	RNA	no	0	[115]
		Dhori virus	RNA	no	0	[116]
		Kadam virus	RNA	no	0	[66,117]
		African horse sickness virus	no	no	2, 4	[118]
		Quaranfil virus	RNA	no	0	[119]
		Sindbis virus	RNA	no	0	[66]
		*Coxiella burnetii*	DNA	no	0	[101,120,121,122,123,124]
		*Francisella persica*	DNA	no	0	[125]
		*Rickettsia aeschlimannii*	DNA	no	0	[121,126,127]
		*Rickettsia africae*	DNA	no	0	[128]
		*Anaplasma* spp./*Ehrlichia* spp.	DNA	no	0	[101]
		*Bartonella bovis et Bartonella rochalimae*	DNA	no	0	[129]
*H. excavatum*	149/211	*Theileria equi*	DNA, pathogen	no	2, 4	[34,130,131]
		*Babesia bigemina*	DNA	no	0	[41]
		*Babesia bovis*	DNA	no	0	[132]
		*Babesia occultans*	DNA	no	3	[30,132]
		*Theileria annulata*	DNA	uncertain	2, 4	[31,41,51,76,78,132,133,134]
		*Theileria capreoli*	DNA	no	0	[31]
		*Theileria ovis*	DNA	no	0	[40,135]
		*Borrelia* spp.	DNA	no	0	[136]
		*Coxiella burnetii*	DNA	no	0	[121,124,137,138,139]
		*Rickettsia africae*	DNA	no	0	[140]
		*Rickettsia aeschlimannii*	DNA	no	0	[140]
		*Anaplasma marginale*	DNA	yes	4	[141]
		*Anaplasma centrale*	DNA	yes	0	[141]
		*Ehrlichia ruminantium*	DNA	no	0	[41]
		*Rickettsia sibirica mongolotimonae*	DNA	no	0	[142]
		CCHFv	RNA, antigen	uncertain	0	[59,61,143]
*H. hussaini*	4/7	*Coxiella burnetii*	DNA	no	0	[144]
		*Rickettsia massiliae, Rickettsia* spp.	DNA	no	0	[38]
*H. impeltatum*	62/88	*Theileria annulata*	DNA	no	2, 4	[41,108,112,145]
		*Theileria lestoquardi (Theileria hirci)*	no	uncertain	0	[146]
		*Theileria ovis*	DNA	no	0	[35]
		*Babesia occultans*	DNA	no	0	[101]
		*Babesia bigemina*	DNA	no	0	[41]
		*Babesia bovis*	DNA	no	0	[41]
		*Babesia pecorum*	DNA	no	0	[35]
		CCHFv	RNA, antigen virus isolation	yes	1, 2, 4 cofeeding	[61,113,147,148,149]
		*Coxiella burnetii*	DNA	no	0	[123,124]
		Alphavirus	RNA	no	0	[66]
		Dhori virus	RNA	no	0	[66]
		Sindbis virus	RNA	no	0	[66]
		*Rickettsia africae*	DNA	no	0	[140]
		*Rickettsia aeschlimannii*	DNA	no	0	[121,150,151]
*H. impressum*	10/17	CCHFv	antigen	uncertain	0	[64]
		*Theileria annulata*	DNA	no	0	[108]
		*Anaplasma/Ehrlichia* spp.	DNA	no	0	[101]
		*Rickettsia africae*	DNA	no	0	[152]
*H. isaaci*	5/5	Kyasanur forest virus	RNA	-	2, 4	[153]
*H. lusitanicum*	68/83	*Theileria equi*	pathogen	no	1, 2, 4	[99,100]
		*Babesia pecorum*	No	yes	0	[154]
		*Theileria annulata*	No	yes	4	[107,155,156]
		CCHFv	RNA, antigen	yes	0	[157,158]
		*Anaplasma phagocytophilum*	DNA	no	0	[159]
		*Borrelia burgdorferi*	DNA	no	0	[160]
		*Borrelia lusitaniae*	DNA	no	0	[161]
		*Coxiella burnetii*	DNA	no	0	[162,163,164,165]
*H. marginatum*	451/620	*Theileria equi*	DNA	yes	0	[130,131,166,167]
		*Theileria annulata*	DNA	yes	2	[41,51,111]
		*Theileria sergenti/orientalis/buffeli*	DNA	no	0	[159,166,168]
		*Theileria ovis*	DNA	no	0	[40,49,169]
		*Theileria lestoquardi*	DNA	no	0	[170]
		*Babesia ovis*	DNA, pathogen in ticks	no	1	[55,171]
		*Babesia caballi*	DNA	yes	0	[130,131,169,172]
		*Babesia bigemina*	DNA	no	0	[41,167]
		*Babesia bovis*	DNA	no	0	[41,167]
		*Babesia occultans*	DNA	yes	2, 3	[30,31,130,134,136,166,173,174,175,176]
		*Babesia microti*	DNA	no	0	[174]
		*Babesia sp. Tavsan1*	DNA	no	0	[31]
		CCHFv	RNA, antigen	yes	1, 2, 3, 4	[64,65,143,158,177,178,179,180,181,182,183,184,185,186,187,188]
		Flavivirus	RNA	no	0	[189]
		Phlebovirus	RNA	no	0	[190]
		Bahig virus	RNA	no	0	[191]
		Batken virus (close to Dhori virus)	RNA	no	0	[192]
		Bhanja virus	RNA	no	0	[193]
		Dhori virus	RNA	no	0	[194,195]
		Tick Borne Encephalitis virus	RNA	no	0	[196]
		Jingmen virus	RNA, virus isolation	yes	0	
						[197]
		Matruh virus	RNA	no	0	[198]
		Tamdy virus	RNA	no	0	[89]
		Wanowrie virus	RNA	yes	0	[153,199]
		West Nile virus	RNA	no	2, 3	[189,200,201,202]
		*Rickettsia aeschlimannii*	DNA	yes	3	[31,136,151,203,204,205,206,207,208]
		*Rickettsia sibirica mongolitimonae*	DNA	no	0	[31]
		*Anaplasma marginale*	DNA	no	0	[31,209]
		*Rickettsia africae*	DNA	no	0	[210]
		*Anaplasma phagocytophilum*	DNA	no	0	[204,211]
		*Anaplasma platys*	DNA	no	0	[211]
		*Coxiella burnetii*	DNA	no	0	[139,142,211,212,213]
		*Francisella tularensis*	DNA	no	0	[214]
		*Ehrlichia monacensis (minasensis)*	DNA	no	0	[204]
		*Ehrlichia ruminantium*	DNA	no	0	[41]
		*Bartonella* spp.	DNA	no	0	[205,213]
		*Borrelia burgdoferi* s.l.	DNA	no	0	[212,213]
		*Borrelia* spp.	DNA	no	0	[136,152]
*H. rufipes*	189/238	*Babesia occultans*	DNA, pathogen	no	2, 3, 4	[175,176]
		*Theileria ovis*	DNA	no	0	[40]
		*Theileria annulata*	DNA	no	2, 4	[108,109,215]
		CCHFv	RNA, antigen, viral particles	yes	1, 2, 3, 4	[64,216,217,218,219,220,221,222,223,224,225]
		Flavivirus	RNA	no	0	[189]
		Dugbe virus	RNA	no	2	[226]
		Alkhurma hemorrhagic fever virus	RNA	no	0	[227]
		St Croix River like virus	RNA	no	0	[228]
		*Rickettsia aeschlimannii*	DNA	no	0	[35]
		*Rickettsia conorii*	DNA	no	0	[229]
		*Anaplasma marginale, centrale, platys*	DNA	no	0	[230]
		*Coxiella burnetii*	DNA	no	0	[74,212,231,232]
		*Borrelia burgdorferi*	RNA and DNA	no	0	[212,233]
*H. schulzei*	17/27	Dhori virus	RNA	no	0	[66]
*H. scupense*	34/47 *H. detritum*: 61/90	*Theileria equi*	No	no	4	[234,235]
		*Theileria annulata*	No	yes	2, 4	[107,236,237,238]
		*Babesia ovis*	DNA	no	0	[40]
		*Theileria ovis*	DNA	no	0	[40]
		*Rickettsia aeschlimannii*	DNA	no	0	[14,205]
		*Rickettsia slovaca*	DNA	no	0	[205]
		*Anaplasma phagocytophilum*	DNA	no	0	[205]
		*Coxiella burnetii*	DNA	no	2, 3	[139,239]
		CCHFv	RNA	uncertain	0	[83]
*H. somalicum*	2/2	*R. conorii*	DNA	no	0	[14]
*H. truncatum*	142/193	*Theileria equi*	DNA	no	0	[101]
		*Babesia caballi*	No	no	3, 4	[240,241]
		*Theileria annulata*	DNA	no	0	[108]
		CCHFv	RNA, viral particles	yes	1, 2, 3, 4, 5 Cofeeding	[113,148,149,217,218,220,222,242,243,244,245,246]
		Bunyamwera virus	RNA	no	0	[247]
		Dugbe virus	RNA	no	0	[248]
		Venezuelan Equine Encephalitis Virus		no	2, 4	[249]
		Kupe virus	RNA	no	0	[248]
		Semliki forest virus	RNA	no	0	[247]
		*Coxiella burnetii*	DNA	no	0	[152,232,250]
		*Borrelia* spp.	DNA	no	0	[152,232]
*H. turanicum*	24/36	CCHFv	RNA	no	0	[187]
		*Rickettsia sibirica mongolitimonae*	DNA	no	0	[140]

CCHFv: Crimean–Congo Hemorrhagic Fever Virus; * epidemiological arguments of possible transmission: for example co-occurrence of a given pathogen in a tick species and in infested vertebrate hosts of the same area, host co-infection with pathogens known to be transmitted by ticks, or the onset of a disease as a result of tick bites: YES, NO, uncertain. ** Experimental validation. 0: none; 1: pathogen reproduction/replication success in ticks; 2: transstadial transmission; 3: transovarial transmission; 4: transmission to a vertebrate host via a tick bite; 5: sexual transmission between male and female ticks.

## Data Availability

The datasets generated during and/or analyzed during the current study can be find in the main text.

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
