# Peer review of "An Update of Evidence for Pathogen Transmission by Ticks of the Genus Hyalomma"

_pathogens, 2023, doi:10.3390/pathogens12040513_

Round 1

Reviewer 1 Report

Dear Authors,

I completed my review on the manuscript entitled “An update of evidence for pathogen transmission by ticks of the genus Hyalomma” by Bonnet et al. I find that the manuscript is well written and organised and the general idea of putting together such type and amount of data is of great interest. Please find below my general observation:

1.     In my opinion the authors should add a reference for the following state: “The detection of 68 mRNA, a priori reflecting the presence of a living organism”

2.     An approximate number should be added instead of “many” in this situation: This is of concern as these ticks are vectors of many pathogens responsible 88 for human and animal diseases.

3.     More details should be given regarding the research strategies in the two database such as: what were limiters/filters used? The number of references included in the table are the initial generated numbers with or without limitations of years/ language etc. All the references were accessible for the authors in the extended format?

4.     In my opinion at least two other important information should be added to the current table: a) the type of evidence is every included article b) For the “Detection of DNA/RNA/antigen/pathogen in ticks” it would be useful to know whether the ticks were engorged or unfed

5.      In case of “pathogen/tick associations for which transmission from an infected host to an initially 131 healthy host via tick bite has been experimentally validated” I suggest highlight in the Table 1 the references that bring the evidence for that. Like this the source for each association will be easily accessible.

6.     The separator for column “Experimental validation of transmission” should be standardized “,” or ”;”.

Author Response

I completed my review on the manuscript entitled “An update of evidence for pathogen transmission by ticks of the genus Hyalomma” by Bonnet et al. I find that the manuscript is well written and organised and the general idea of putting together such type and amount of data is of great interest. Please find below my general observation:

  1. In my opinion the authors should add a reference for the following state: “The detection of 68 mRNA, a priori reflecting the presence of a living organism”

This fact is so widely recognized and known that it is difficult to find an ad hoc reference. See the definition of RNA from NHI for example: “Ribonucleic acid (abbreviated RNA) is a nucleic acid present in all living cells that has structural similarities to DNA.“

  1. An approximate number should be added instead of “many” in this situation: This is of concern as these ticks are vectors of many pathogens responsible 88 for human and animal diseases.

We cannot mention a number here because, and this is precisely the point of this paper, the levels of proof of transmission are extremely variable according to the pathogens concerned. The list of the concerned pathogens is presented in the Table

  1. More details should be given regarding the research strategies in the two database such as: what were limiters/filters used? The number of references included in the table are the initial generated numbers with or without limitations of years/ language etc. All the references were accessible for the authors in the extended format?

Thanks for this suggestion. This was added in the new version of the MS and we hope that it is now OK.

  1. In my opinion at least two other important information should be added to the current table: a) the type of evidence is every included article b) For the “Detection of DNA/RNA/antigen/pathogen in ticks” it would be useful to know whether the ticks were engorged or unfed
  2. In case of “pathogen/tick associations for which transmission from an infected host to an initially 131 healthy host via tick bite has been experimentally validated” I suggest highlight in the Table 1 the references that bring the evidence for that. Like this the source for each association will be easily accessible.

If we understand well these two comments 4 &5 correctly, the information requested (type of evidence, level of engorgement) can all be found in the references provided. Although we agree with the reviewer, adding this information to the table by separating the ad hoc references would greatly increase the size and complexity of the table which are already very consistent and we do not want to do so.

  1. The separator for column “Experimental validation of transmission” should be standardized “,” or ”;”.

It was corrected

Reviewer 2 Report

The authors provide an exhaustive revision related to pathogen transmission by ticks of the genus Hyalomma and conclude that there are few validated cases of vector competence.

Minor comments

None of the scientific names in the document are in italic font - please correct throughout the manuscript.

Line 30 - replace "to collate the validation" with "to validate the"

Include more references for the statement on lines 53 to 58, as the reference included only supports the co-feeding transmission.

The document, even if it is not formerly separated in sections should include a couple of paragraphs of results and discussion, describing and analyzing the results from the table. Please include 1 to 2 paragraphs of results and discussion after line 127.  

Author Response

The authors provide an exhaustive revision related to pathogen transmission by ticks of the genus Hyalomma and conclude that there are few validated cases of vector competence.

Minor comments

None of the scientific names in the document are in italic font - please correct throughout the manuscript.

There must have been a problem at the time of download because the paper had been submitted with the Latin names in italics

Line 30 - replace "to collate the validation" with "to validate the"

We are sorry to disagree but the meaning would not be the same if we made this change

Include more references for the statement on lines 53 to 58, as the reference included only supports the co-feeding transmission.

Thank you for that comment. The MS already contains a very large number of references, so we have chosen to mention a general review here.

The document, even if it is not formerly separated in sections should include a couple of paragraphs of results and discussion, describing and analyzing the results from the table. Please include 1 to 2 paragraphs of results and discussion after line 127.  

The unusual format of the manuscript which does not include the classical sections requested by the reviewer was proposed to the newspaper in agreement with the editor. Thus, we want to design this format in a simple table with a comment without discussion.